# Hops across Continents: Exploring How Terroir Transforms the Aromatic Profiles of Five Hop (*Humulus lupulus*) Varieties Grown in Their Countries of Origin and in Brazil

**DOI:** 10.3390/plants13192675

**Published:** 2024-09-24

**Authors:** Marcos Edgar Herkenhoff, Oliver Brödel, Marcus Frohme

**Affiliations:** 1Department of Biochemical and Pharmaceutical Technology, School of Pharmaceutical Sciences, University of São Paulo (USP), Av. Professor Lineu Prestes, 580, São Paulo 05508-000, SP, Brazil; 2Food Research Center FoRC, University of São Paulo (USP), Av. Professor Lineu Prestes, 580, São Paulo 05508-000, SP, Brazil; 3Division Molecular Biotechnology and Functional Genomics, Technical University of Applied Sciences Wildau, 15745 Wildau, Germany; oliver.broedel@th-wildau.de (O.B.); mfrohme@th-wildau.de (M.F.)

**Keywords:** *Humulus lupulus*, aromatic profile, terroir impact, volatile compounds, HS-SPME/GC-MS

## Abstract

*Humulus lupulus*, or hops, is a vital ingredient in brewing, contributing bitterness, flavor, and aroma. The female plants produce strobiles rich in essential oils and acids, along with bioactive compounds like polyphenols, humulene, and myrcene, which offer health benefits. This study examined the aromatic profiles of five hop varieties grown in Brazil versus their countries of origin. Fifty grams of pelletized hops from each strain were collected and analyzed using HS-SPME/GC-MS to identify volatile compounds, followed by statistical analysis with PLS-DA and ANOVA. The study identified 330 volatile compounds and found significant aromatic differences among hops from different regions. For instance, H. Mittelfrüher grown in Brazil has a fruity and herbaceous profile, while the German-grown variety is more herbal and spicy. Similar variations were noted in the Magnum, Nugget, Saaz, and Sorachi Ace varieties. The findings underscore the impact of terroir on hop aromatic profiles, with Brazilian-grown hops displaying distinct profiles compared to their counterparts from their countries of origin, including variations in aromatic notes and α-acid content.

## 1. Introduction

*Humulus lupulus*, commonly known as hops, is a species of flowering plant in the Cannabaceae family. It is native to Europe, Asia, and North America and is primarily known for its use in brewing beer [1]. The plant is a vigorous, climbing vine with rough stems and serrated leaves arranged oppositely along the stem. Hops are dioecious, meaning there are separate male and female plants. The female plants produce cone-like structures called strobiles, which are used in brewing to impart bitterness, flavor, and aroma to beer. These cones contain lupulin glands, which contain the essential oils and acids responsible for the characteristic bitterness and aroma of hops [2].

Apart from the most common compounds found in hop cones belonging to bitter acids (α- and β-acids) [3], there are at least several other bioactive compounds (essential oils and polyphenols) that make hop cones a feedstock with a broad range of microbiological properties [2,3,4]. Among various properties, hop cones contain compounds, such as prenylated flavonoids, which have been shown to possess sedative properties [5]. Certain compounds found in hops, such as phytoestrogens, have been investigated for their potential in hormone regulation. These compounds may have implications for conditions such as menopausal symptoms [6].

Hops offer health benefits due to their antioxidants, like polyphenols, which may reduce oxidative stress and chronic disease risk [7]. Compounds such as humulene and myrcene in hops are believed to have relaxing effects [8,9]. In addition, hundreds of aroma compounds are found in hop essential oils [10], though these oils constitute only about 0.5% to 3.0% of the hops’ dry weight [3]. The complex composition of hop essential oil makes characterizing its aroma a challenging task.

Despite the importance of characterizing aroma-related compounds in hops, the extraction methodologies commonly employed are often not very effective. Extraction techniques include steam distillation (SD), simultaneous distillation extraction (SDE), direct solvent extraction (DSE), and solvent-assisted flavor evaporation (SAFE). While SD and SDE are conventional methods, they can decompose volatile compounds due to high temperatures [11]. DSE extracts both volatiles and non-volatiles but is often used in combination with SAFE for thorough isolation. Although DSE-SAFE is an effective method, its high equipment costs and complexity limit accessibility. Headspace solid-phase microextraction (HS-SPME) is preferred for its solvent-free extraction and minimal sample volume requirements [11]. 

Regarding volatile and aromatic compounds, Su and Yin [11] conducted a study aimed at analyzing five fresh samples of Cascade and Chinook hops from different locations in Virginia using headspace solid-phase microextraction gas chromatography mass spectrometry olfactometry (HS-SPME-GC-MS-O). They identified 33 aromatic compounds, including esters, monoterpenes, sesquiterpenes, terpenoids, an aldehyde, and an alcohol. Furthermore, the authors demonstrated how the cultivation location can significantly influence the aroma profiles of Cascade and Chinook hops [11]. This study demonstrated the effect of location on the production of volatile compounds in these hop varieties. Additionally, in Brazil, there has been an expansion in hop production involving foreign varieties.

Based on these principles and considering that volatile compounds are extremely important for determining the hop profile, as well as their application in the food, cosmetics, or pharmaceutical industries, this study aimed to evaluate the aromatic profile of the hop varieties Hallertauer Mittelfrüher, Magnum, Nugget, Saaz, and Sorachi Ace. Except for Sorachi Ace, the study compared samples from their countries of origin with samples of the same varieties grown in Brazil using headspace solid-phase microextraction coupled with gas chromatography–mass spectrometry (HS-SPME/GC-MS).

## 2. Material and Methods

### 2.1. Samples

For this study, 50 g samples of pelletized hops from five distinct strains were collected, with samples planted in their countries of origin, except for Sorachi Ace, which was compared with samples of the same strains planted in Brazil. Samples of Magnum and Hallertauer Mittelfrüher hops were obtained from Germany, Nugget and Sorachi Ace from the United States, and Saaz from the Czech Republic, all sourced from Barth Haas (Nuremberg, Germany). The Brazilian hops, Hallertauer Mittelfrüher, Nugget, and Saaz, were sourced from Dalcin (Taguaí, SP, Brazil), and the Magnum and Sorachi Ace hops from Brava Terra (Fortuna, SP, Brazil) (Table 1).

The samples were manually ground into a fine powder using a mortar and pestle for subsequent analysis. Ground hop samples (40 ± 0.5 mg) were placed in a 20 mL glass vial with an automatic sampler. The vials were sealed with PTFE/silicone septa and aluminum caps (Macherey-Nagel, Bethlehem, PA, USA).

### 2.2. Instrumentation

The volatile compound profiles were analyzed using headspace solid-phase microextraction (HS-SPME) combined with gas chromatography–mass spectrometry (GC–MS). This analysis employed the GCMSQP2020 NX system, incorporating the Nexis GC-2030 gas chromatograph, a quadrupole mass spectrometer, and the AOC-6000 Plus autosampler, all supplied by Shimadzu (Nakagyo-ku, Kyoto, Japan). For HS-SPME extraction, a DVB/CAR/PDMS (divinylbenzene–carboxen–polydimethylsiloxane) Smart Fiber (80 μm) from Shimadzu was utilized.

Prior to analysis, the fiber was preconditioned at 240 °C, and two blank injections were performed according to the manufacturer’s guidelines. Samples were equilibrated for 10 min at 50 °C in the autosampler’s heat block. The extraction process involved exposing the SPME fiber to the sample headspace for 50 min. The fiber was then inserted into the GC injector port for 3 min at 230 °C in splitless mode (using an SPME glass liner with a 0.75 mm ID), enabling thermal desorption of the volatile compounds. GC separation was performed with a constant helium flow (1 mL/min) on a PEG capillary column (HP-INNOWAX, 30 m, 0.25 mm ID, 0.15 μm) from Shimadzu. The oven temperature was programmed to increase from 40 °C to 150 °C at a rate of 5 °C per minute, followed by a ramp to 225 °C at 20 °C per minute, with initial and final holding times of 5 min and 20 min, respectively, as described by Su and Yin [11].

Mass spectrometry detection was carried out using electron-impact (EI) ionization at 70 eV in full-scan mode within the 40–350 amu range. The transfer line and ion source were maintained at 250 °C. Data acquisition was conducted in total ion current (TIC) mode.

### 2.3. Volatile Compounds Identification and Database Software Analysis

The detection of volatile compounds was conducted by comparing each peak’s molecular fragmentation pattern against the mass spectra available in the 2020 NIST MS database library (National Institute of Standards and Technology, Gaithersburg, MD, USA). A compound was considered identified if it displayed a similarity index (SI) exceeding 85. In cases of ambiguous identifications, retention indices were calculated using a series of n-alkanes (C8–C23) as references for confirmation.

Chromatographic profiles from the samples were analyzed using chemometric classification methods. These methods aim to leverage experimental data to predict the qualitative properties of the samples, referred to as categories or classes. Specifically, the goal was to determine the aroma characteristics of the hop samples. Given the multivariate nature of the experimental data (i.e., the chromatographic profiles), this study focused on employing partial least squares discriminant analysis (PLS-DA) to construct a classification model.

Each identified compound was queried using its CAS Registry Number in the PubChem database (https://pubchem.ncbi.nlm.nih.gov/) (accessed on 1 August 2024). Furthermore, the flavor and aroma profiles of these compounds were examined using the Perflavory database (https://perflavory.com/search.php) (accessed on 1 August 2024).

### 2.4. Statistical Analysis

For statistical analysis, the results were presented as mean ± standard deviation (SD). To quantify the volatile compounds common in at least two of the analyzed styles, the identified peak areas were automatically converted into Area% using LabSolutions GCMSolutions software version 1 (Shimadzu, Kyoto, Japan). This quantification approach was adopted because, as per the manufacturer, employing a specific standard for quantification ensures consistent concentration levels across all samples. Student’s *t*-test was utilized for comparing two samples, given that normal distribution was confirmed. For comparing three or more samples, an analysis of variance (ANOVA) was performed, followed by the Tukey test. Differences were considered statistically significant when *p* ≤ 0.05 (5% significance level).

## 3. Results and Discussion

### 3.1. Classification by PLS-DA

To investigate data trends and sample correlations, a multivariate analysis was utilized. A classification model was specifically created to highlight the differences related to the production method. Chromatographic profiles, illustrated as GC-MS total ion currents (TIC), were processed using PLS-DA to distinguish between the hop varieties planted in Brazil and those in their countries of origin (Figure 1).

To evaluate chemical differences between the beer groups analyzed, variable importance in projection (VIP) scores were calculated from the PLS-DA model. VIP scores measure the contribution of individual variables to the model, with higher scores indicating greater importance. Normalized VIP scores greater than one are generally considered significant. By combining PLS regression coefficients with VIP scores, we can identify key compounds for distinguishing among sample types and gain insights into the direction of observed variations.

### 3.2. Aromatic Hop Profile

A total of 330 different volatile compounds were identified using HS-SPME/GC-MS across all hop samples (Appendix A). Although several of these compounds exhibit distinct aroma and odor profiles, the suppliers of these hops, as well as Beer Maverick (https://beermaverick.com/) (accessed on 1 August 2024), have already reported differentiated profiles (Figure 2).

H. Mittelfrüher grown in Germany has a more spicy and herbal profile compared to the one grown in Brazil, which has a greener profile (Figure 2). Magnum grown in Germany has a spicier profile, whereas the one grown in Brazil features floral, berry, tropical fruit, citrus, and herbal notes. Nugget grown in the United States has a more herbal and woody profile, while the same variety grown in Brazil presents citrus, floral, and berry characteristics. Saaz grown in the Czech Republic has a slightly more woody and floral profile, while the same variety in Brazil exhibits more herbal, spicy, and citrus notes. Finally, Sorachi Ace grown in the United States shows almost the same pattern as the same variety in Brazil, except that the Brazilian variety is slightly more woody, tropical, citrus, herbal, and floral but maintains a very similar sensory pattern. It is interesting to note that the same variety planted in its country of origin, in this case, Japan, and according to Beer Maverick (https://beermaverick.com/), also has the same profile, except for the absence of a woody profile. This may indicate a variety with little terroir effect.

A study analyzed 33 active aromatic compounds in hop samples from the Cascade and Chinook varieties harvested from different locations in Virginia. Using chromatography and olfactometry techniques, the presence of esters, monoterpenes, sesquiterpenes, and terpenoids, among other compounds, was identified, exhibiting various aromatic characteristics, such as fruity, herbal, woody, and citrus notes [11].

### 3.3. Hallertauer Mittelfrüher

H. Mittelfrüher (HM) is a hop variety that has shown a higher α-acid content when planted in Brazil, according to its suppliers (Table 1). When grown in Taguaí, São Paulo, it presents a content of 6.88%, compared to 4.50% in Germany. A more recent study with the same hop variety from the western region of Paraná, Brazil, indicated α-acid levels of 5.9%, β-acid levels of 1.80%, and an essential oil content of 1.1 mL/100 g [12].

In terms of compound quantities, calculated by % area, 14 compounds were more expressed in the German HM, while 23 were more expressed in the Brazilian HM (Table 2). Among the 14 compounds more expressed in the German variety, 11 are related to aroma or odor. In the Brazilian HM, 13 of the 23 more-expressed compounds are associated with aroma or odor. Regarding unique compounds, 64 were identified in the German HM compared to Brazilian HM, with 28 of these related to aroma (Table 3). In the Brazilian HM, 45 unique compounds were found compared to the German HM, with 22 being related to aroma.

The German HM exhibits an aromatic profile rich in herbal, spicy, floral, fruity, and woody notes, whereas those from Brazil have an aromatic profile characterized by fruity, herbal, sweet, and woody notes. This aromatic profile aligns with the descriptions provided by the suppliers for both (Figure 2).

Among the compounds that are more highly expressed, we can highlight dextro-limonene (citrus), also known as D-limonene (Table 2). This compound acts against the cytoplasmic membranes of microorganisms, resulting in a loss of membrane integrity, altering its permeability, and leading to the loss of ions and proteins [13]. Benzaldehyde (fruity) is another significant compound, known as an inhibitor of quorum sensing for the opportunistic pathogen Pseudomonas aeruginosa [14]. Hexanoic acid (fatty) has been considered to have high sensory potential effects in Chinese ‘Marselan’ wines [15]. Additionally, 2-undecanone (fruity) is known for its antifungal activity against *Colletotrichum gloesporioides* [16]. Another property of this compound is its ability to alleviate asthma by reducing airway inflammation and remodeling. This beneficial effect is achieved through the inhibition of the NF-κB pathway [17].

Among the unique compounds in the Brazilian HM (Table 3), notable ones include 1-octen-3-ol (earthy), known for its antioxidant and antimicrobial properties [18]. This compound acts as a defense mechanism in seaweeds [19], potentially enhancing food preservation and contributing to overall health. It is also associated with aging flavors [18]. 1-Hexanol (herbal) is found abundantly in Pale Ale and Lambic beer styles [20] and holds significant potential for applications in the food and cosmetic industries [21]. Ethyl hexanoate (fruity), providing flavors typical of apples and pineapples, is a maturity marker in pequi fruits (*Caryocar brasiliense*) and is the most predominant compound in this fruit [22]. Methyl isobutyrate (fruity) can be detected in numerous foods and beverages and has been identified as a key volatile compound in Hunan Changde rice noodles fermented with *Lactococcus* [23]. Hexyl acetate is frequently used as a flavoring agent in a variety of food products, including candies, baked goods, and beverages. It is also an ingredient in perfumes, soaps, and other personal care products [24]. Moreover, the hexyl acetate identified in the grape pomace of the investigated grape varieties can be used similarly, serving as a flavoring agent in various food items and as a component in perfumes, soaps, and other personal care products [21]. This ester is also known for imparting a fragrance known as ‘Orange Beauty’ [25].

Among the most expressed compounds in the Brazilian HM, noteworthy ones include methyl heptanoate (fruity), which contributes to a fruity flavor and is found in various fruits. Methyl octanoate (waxy) adds a smooth, sweet flavor, common in some fruits and wines, and is one of the main flavoring agents in foods, possessing a vinous, fruity, and orange-like odor [26]. Octanol (waxy) has a pleasant aroma that contributes to the complexity of flavors in foods and is common in various beer styles [20].

### 3.4. Magnum

Unlike H. Mittelfrüher, Magnum is a hop variety that showed a lower α-acid content when planted in Brazil, according to its suppliers (Table 1). Regarding the quantity of compounds, calculated by % area, 26 compounds were more expressed in the German Magnum, while 20 were more expressed in the variety planted in Brazil (Table 2). Of the 26 more expressed in the German Magnum, 17 are related to aroma or odor. Of the 23 compounds more expressed in the Brazilian Magnum, 11 are related to aroma or odor. Regarding the compounds found in the German Magnum, 34 unique compounds were identified compared to the Brazilian Magnum (Table 3). Of this total, only 22 are related to aroma. In the Brazilian Magnum, seven unique compounds were found compared to the German Magnum. Among them, six are related to aroma.

The German Magnum hop profile is characterized by a blend of herbal, terpenic, woody, citrus, fruity, floral, and fatty notes. The key compounds contributing to this profile include alpha-pinene (herbal), alpha-phellandrene (terpenic), and geranyl butyrate (fruity), among others. The Brazilian Magnum exhibits a profile with fruity, floral, and woody notes. The key compounds responsible for this profile include 2-methylpropyl 3-methylbutanoate (fruity), methyl heptanoate (fruity), and (+)-delta-cadinene (herbal). The aromatic profiles of the German and Brazilian Magnum hop varieties are influenced by the specific compounds present in each group. The German Magnum is characterized by a complex mix of herbal, fruity, and woody notes, while the Brazilian Magnum is dominated by fruity, floral, and herbal aromas. These differences are due to the unique combination of compounds present in each group, influenced by factors such as terroir and cultivation practices.

Beer flavored with total Magnum hop oil has a unique sensory profile, featuring strong “crushed grass, sap”, “resinous”, “earthy”, and “musty” aromas. Magnum hop oil consists mainly of β-myrcene, β-caryophyllene, and α-humulene, making up to 80% of the oil. β-myrcene, the most abundant compound, can smell “spicy”, “resinous”, “metallic”, or “geranium-like” at different concentrations. The aromas of β-caryophyllene and α-humulene are less distinct but are described as “rubber-like”, “mouldy”, “woody”, and “spicy.” These aroma characteristics are also found in sesquiterpene-flavored beer, but at lower intensities, indicating that the total oil enhances them [27].

### 3.5. Nugget

In contrast to the hops mentioned in the previous sections, the α-acid content was the same in both the American and Brazilian Nugget (Table 1). Regarding the quantity of compounds, calculated by % area, 25 compounds were more expressed in the variety planted in the United States, while 19 were more expressed in the variety planted in Brazil (Table 2).

Among the 25 compounds more expressed in the American Nugget, 17 are related to aroma or odor (Table 2). Among the 19 compounds more expressed in the Brazilian Nugget, 7 are related to aroma or odor. Regarding the compounds found in the variety planted in the United States, 42 unique compounds were identified compared to the Brazilian Nugget (Table 3). Of this total, only 25 are related to aroma. In the Brazilian Nugget, 37 unique compounds were found compared to the American Nugget. Among them, 24 presented some relation to aroma.

Among the compounds most abundantly expressed in the American Nugget, notable ones include dextro-limonene (citrus), which is one of the primary compounds identified in the essential oil of Okoume (*Aucoumea klaineana*) [28], being studied for its bioactive constituents and antibacterial activities. Linalool (floral) is commonly found in large quantities in Indian Pale Ale beers but in smaller amounts in Pale Ale (IPA) [20]. 3-Methylbutyl 2-methylbutanoate (fruity) is reported to occur in foods such as apple brandy (Calvados), banana (*Musa sapientum* L.), cider (apple wine), date (*Phoenix dactylifera* L.), grape brandy, and passion fruit (Passiflora species), among others [29].

Among the compounds most abundantly expressed in the variety planted in Brazil, noteworthy is methyl octanoate (waxy), which, interestingly, was found in Annona crassiflora, known as marolo fruit from the cerrado biome. This species is one of the most consumed in the Brazilian Midwest, with this compound significantly contributing to its aroma [30].

Among the compounds found exclusively in the American Nugget, methyl isobutyrate (fruity) was previously mentioned as a volatile compound in Hunan Changde rice noodles fermented by Lactococcus [23] and is also an exclusive compound of Brazilian HM hops. Dimethyl disulfide (sulfurous) adds a sulfurous character, which is interesting for certain aromatic profiles. It is also one of the most abundant compounds in microwave-cooked radish (*Raphanus sativus* L.) oils [31]. Isovaleric acid (cheesy) imparts a cheesy character and was exclusively found in Bock beers [20], being perceived by many individuals as a very strong odor impression [32].

Among the compounds found exclusively in the Brazilian Nugget, hexyl acetate (fruity) is also an exclusive compound of Brazilian HM hops. It is a flavoring agent in a variety of food products and an ingredient in perfumes, soaps, and other personal care products [24]. Methyl dodecanoate (waxy) adds a waxy and sweet flavor and is one of the most variable compounds for distinguishing wine cultivars, contributing significantly to their sensory characteristics [33]. It was also isolated from the ethyl acetate extract of the culture filtrate of the probiotic Lactobacillus plantarum H24 [34]. 3-methylbutyl 3-methylbutanoate (fruity) contributes a fruity flavor, enhancing the complexity of the aroma. It is the most abundant compound in all of the flowering stages of Asian skunk cabbage (*Symplocarpus renifolius*, Araceae) [35] and is present in the odors of ripe bananas, guavas, and oranges. It is also found among the compounds that attract both sexes of the invasive African fruit fly *Bactrocera invadens* [36].

### 3.6. Saaz

The Saaz hop exhibits an α-acid content of 3.5 for the Czech, compared to 5.67 for the Brazilian (Table 1). Regarding the quantity of compounds, calculated by % area, 17 compounds were more expressed in the Czech Saaz, while 15 were more expressed in the Brazilian Saaz (Table 2).

Among the 17 compounds more expressed in the Czech Saaz, 10 are related to aroma or odor (Table 2). Of the 15 compounds more expressed in the Brazilian Saaz, 9 are related to aroma or odor. Regarding the compounds found in the variety planted in the Czech Saaz, 26 unique compounds were identified compared to the Brazilian Saaz (Table 3). Of this total, only 18 are related to aroma. In the Brazilian Saaz, 28 unique compounds were found compared to the Czech Saaz. Among them, 18 were related to aroma.

The differences in the aromatic profiles of the Czech and Brazilian Saaz are largely due to the variation in the expression of specific compounds, which are influenced by factors such as climate, soil, and cultivation practices. The presence of more spicy, earthy, woody, and floral compounds in the Czech Saaz suggests a more traditional and balanced aroma profile, while the Brazilian Saaz’s higher expressions of ethereal, herbal, fruity, and spicy compounds offer a unique and potentially more vibrant aroma.

These aromatic differences are crucial for brewers when selecting hops for specific beer styles, as the aroma profile of the hops can significantly influence the final product’s flavor and aroma. Understanding the specific compounds responsible for these aromas allows brewers to tailor their hop selections to achieve the desired sensory characteristics in their beers.

Regarding the compounds most expressed in Czech Saaz hops, linalool (floral) is commonly found in higher quantities in Indian Pale Ale beers [20]. 1-Octen-3-ol (earthy), known for its mushroom-like aroma, is a byproduct of the enzymatic degradation of linoleic acid and ethanol [37], and previous studies have shown that it functions as a defense compound in marine algae [19,38,39].

Among the compounds most expressed in Brazilian Saaz hops, benzaldehyde (fruity) primarily manifests as a sweet note. Regarding the unique compounds in Czech Saaz hops, isovaleric acid (cheesy) is noted for its sweaty–cheesy aroma and contributes to the sensory characteristics of Gouda cheese [40]. Hexanoic acid (fatty) plays a role in aromatic complexity and has been documented for its effects in Chinese wines [15].

For the unique compounds in Brazilian Saaz hops, hexyl acetate (fruity) is widely used in the food and cosmetic industries [24], is present in grape pomace [21], and is recognized for its distinct ‘Orange Beauty’ fragrance [25]. 3-Methylbutyl 3-methylbutanoate (fruity) contributes a fruity flavor and is the predominant compound in the flowering of *Symplocarpus renifolius*, Araceae [35].

### 3.7. Sorachi Ace

The Sorachi Ace (SA) hop exhibits an α-acid content of 10.8 in the American, while the Brazilian SA has an α-acid content of 8.7 (Table 1). Regarding the quantity of compounds, calculated by % area, 24 compounds were more expressed in the American SA, whereas 21 were more expressed in the Brazilian SA (Table 2).

Of the 24 compounds more expressed in the American SA, 13 are related to aroma or odor (Table 2). Of the 21 compounds more expressed in the Brazilian SA, 12 are related to aroma or odor. Regarding the compounds found in the American SA, 17 unique compounds were identified compared to the same Brazilian SA (Table 3). Of this total, only 13 are related to aroma.

In the Brazilian SA, 25 unique compounds were found compared to the American SA, among them, 17 have some relation to aroma.

The American SA tends to have a more diverse aromatic profile with herbal, citrus, and woody notes being dominant. The presence of compounds like 2,6,6-trimethylbicyclo[3.1.1]hept-2-ene and dextro-limonene significantly contribute to the herbal and citrus characteristics. The Brazilian SA exhibits a more fruity and spicy aromatic profile. Compounds such as myrcene and 3-methylbut-2-enyl 2-methylpropanoate are responsible for these attributes, providing a unique aroma compared to its American counterpart. The presence and concentration of specific volatile compounds influence the aromatic profile. Terpenes and esters are particularly important, as they are responsible for the distinct aromas of hops, affecting beer’s aroma and flavor. The unique environmental conditions and cultivation practices in the US and Brazil likely contribute to the differences in these aromatic compounds, resulting in variations in the hop profiles.

Sorachi Ace is a hop variety that imparts distinctive varietal aromas to the final beer, including woody, pine, citrus, dill, and lemongrass notes. The research concludes that the unique aroma of Sorachi Ace hops is due to the high levels of geranic acid, which acts synergistically with other hop-derived compounds to enhance the overall aroma. Sensory evaluations showed that late and dry hopping resulted in beers with significantly higher scores for flowery, fruity, tropical, and lemon characteristics compared to kettle-hopped beer [41].

Regarding the compounds most expressed in the American SA, we can highlight dextro-limonene (citrus), which is a compound exclusively found in the Gose style compared to other sours [20]. It is found in the essential oil of okoume [28] and also acts against the cytoplasmic membranes of microorganisms [13]. Benzaldehyde (fruity) provides fruity notes. Linalool (floral) is common in IPA. Regarding the compounds most expressed in the Brazilian SA, we can highlight 3-methylbut-2-enyl 2-methylpropanoate (fruity), which is one of the exclusive compounds found in the Farmhouse Ale beer style. 1-Octen-3-ol (earthy) is known for being an antioxidant and antimicrobial [18], acting as a defense mechanism in marine algae [19], and is associated with the aging of flavors [18]. This compound is produced from 10-linoleate hydroperoxide [42] and is the most abundant alcohol found in soybeans cultivated in North America [43]. 2-Nonanone (fruity) is a bioactive compound capable of promoting rice growth [44]. It was identified in the volatilome of *Bacillus* sp. BCT9, showing the ability to increase lettuce biomass by up to 48% after 10 days of exposure [45]. Regarding the compounds exclusive to the American SA, we can highlight isovaleric acid (cheesy), which, as discussed, is found exclusively in Bock beers [20] and has a strong impression on individuals [32]. Camphene (woody) is a component of rosemary (*Rosmarinus officinalis*) essential oil (Hendel et al., 2024). 1-Hexanol (herbal) is found in Lambic and Pale Ale beers [20] and has potential applications in the food industry (Abreu et al., 2023). Methyl heptenone (citrus) is a compound that signals freshness in algae (*Cladostephus spongiosus*) [46]. *Hannaella* and *Neomicrosphaeropsis* showed a significantly positive correlation with this compound produced during the fermentation of Petit Manseng sweet wine [47].

Regarding the compounds exclusive to the Brazilian SA, we can highlight hexyl acetate (fruity). As discussed, it is an important compound for food and perfume [24]. Octyl acetate (floral) is an exclusive compound found in Pilsen beers with hop extract [20], as well as in the Quadruppel beer style as an exclusive compound [20]. This compound is a useful marker for monitoring the fermentation process, as its post-fermentation concentration increases proportionally to the pre-fermentation concentration of the corresponding alcohol [48]. Large amounts of this compound have also been associated with potential antioxidant and anticancer effects in the leaves of the Pittosporum species (Pittosporaceae) [49]. Pentyl propanoate (fruity), which is a metabolic product derived from 1-pentanol, is an important flavoring ingredient formed by the condensation of pentanol and propanoic acid. The fruity smell of esters makes them unique, with wide applications in the flavor, fragrance, and solvent industries [50]. This compound is also found in truffles (*Tuber canaliculatum*) harvested in Quebec, Canada [51]. Alpha-Phellandrene (terpenic) is a compound found in the industrial product Monash Pouch diet, although its role in the flavor of this product is still unknown [52]. 3-Methylpentanoic acid (animal) receives special attention due to its peculiar aroma and its importance for fermented beverages. Acids can be obtained by lipid oxidation or by the conversion of aldehydes or ketones. Additionally, acids can react with alcohols to form esters and provide wine aroma, among them 3-methylpentanoic acid [53], which is also found in rice wine [54]. The presence of this compound seems to be stable in amounts in different wines [55] and has also been described in beers [56]. It is interesting to note that this aroma is present in the Brazilian SA, which can be very interesting for the country’s scenario. Recently, Brazil developed its first beer style, the Catharina sour. This style has been studied [57,58] and has already shown complexity in its volatile compound composition [20]. Thus, this could be a good hop for the local production of Catharina sour-style beers, even to enhance the flavor.

### 3.8. The Brazilian Touch in Hops

Regarding hops cultivated in Brazil, we observed a pattern, specifically compounds that were present in higher quantities or exclusively in all Brazilian-cultivated varieties. All varieties showed a higher amount of methyl 6-methylheptanoate (unrelated to aroma) and lower amounts of 3-(4-methylpent-3-enyl) furan (woody), linalool (floral), and 2-undecanone (fruity). Concerning exclusive compounds, none were found to be common in all varieties planted in Brazil. However, some are frequent in more than one hop. Pentyl propanoate (fruity), hexyl acetate (fruity), and 2-methylpropyl propanoate (fruity) were found in all hops except H. Mittelfrüher. Geranyl propionate (floral) was found in all except Magnum. Octyl acetate (floral) and octyl propanoate (fruity) were absent only in the German varieties H. Mittelfrüher and Magnum. Cis-3-hexenyl butyrate (green) and Geranyl acetate (floral) were absent only in the varieties H. Mittelfrüher and Sorachi Ace.

Regarding hops cultivated in their places of origin, isovaleric acid (cheesy) was the only exclusive compound found in all varieties. Dimethyl disulfide (sulfurous) was found in all except Sorachi Ace, while acetone (solvent) was found in all except Magnum, and dodecane (alkane) in all except H. Mittelfrüher. Methyl caproate (fruity), alpha-cadinol (herbal), and camphene (woody) were found in all American hops, possibly indicating a terroir effect.

H. Mittelfrüher and Magnum strains, which are German strains, have shown unique compounds compared to the German hop versus the Brazilian ones. Fourteen compounds are common between the H. Mittelfrüher and Magnum strains planted in Germany (Table 3), seven of which are aromatic, namely 2-methyl-1-butanol (ethereal); methyl isobutyrate (fruity); dimethyl disulfide (sulfurous); alpha-phellandrene (terpenic); isovaleric acid (cheesy); heptyl isobutyrate (fruity); and linalool oxide (earthy); while two compounds are common between the H. Mittelfrüher and Magnum strains planted in Brazil, namely alpha-muurolene (herbal) and beta-bisabolene (green).

It is given that H. Mittelfrüher and Magnum are both cultivated in the same region in Germany, known as Hallertau in Bavaria, and our Brazilian hop also originates from the same region. The cultivation in Germany would suggest a hop with more fruity, earthy, and cheesy notes, while in Brazil, it would suggest more herbal and green notes. Of course, further studies would be necessary to confirm this.

## 4. Conclusions

The analysis conducted through PLS-DA revealed significant variations in the aromatic profiles of hops cultivated in Brazil compared to their native counterparts, uncovering a total of 330 distinct volatile compounds. The differences in volatile compounds among the varieties, such as Hallertauer Mittelfrüher, Magnum, Nugget, Saaz, and Sorachi Ace, reflect the influence of terroir and cultivation conditions. Although the α-acid content of Nugget is similar between the Brazilian and American samples, the Magnum hop grown in Brazil stood out for its more fruity and floral profile. The Brazilian Saaz, on the other hand, exhibited an increase in α-acid content and fruity compounds compared to its Czech counterpart. These variations not only enrich the sensory complexity of beers but also open new opportunities for innovations in the industry, allowing brewers to select hops suitable for different styles, thereby enhancing the diversity and quality of the beverages produced.

## Figures and Tables

**Figure 1 plants-13-02675-f001:**
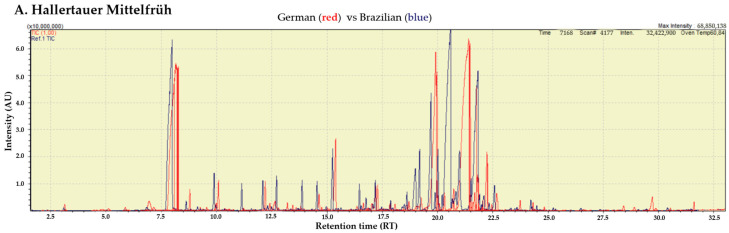
Chemical interpretation of the PLS-DA model discriminating between hops planted in their countries of origin (red) and the same varieties planted in Brazil (blue). Samples are based on VIP scores and regression coefficients. The selected hop varieties were (**A**) Hallertauer Mittelfrüh, (**B**) Magnum, (**C**) Nugget, (**D**) Saaz, and (**E**) Sorachi Ace. The chromatogram regions significantly contribute to the PLS-DA model.

**Figure 2 plants-13-02675-f002:**
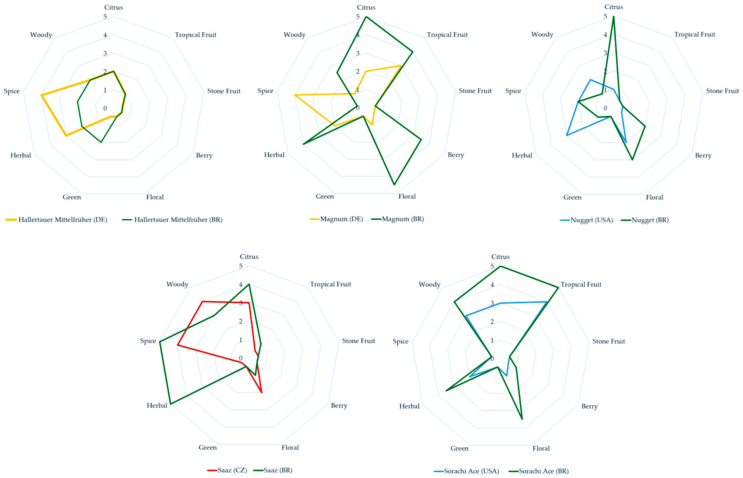
Aromatic profiles according to the hop (*Humulus lupulus*) strain producers for the varieties used in this study grown in their countries of origin (DE: Germany; USA: United States; and CZ: Czech Republic) and grown in Brazil (BR).

**Table 1 plants-13-02675-t001:** Characteristics of the five hop (*Humulus lupulus*) varieties used in the present study.

Hop Strain	Typical Use	Company	Origin	Harvest	Alpha Acid (%) *
Hallertauer Mittelfrüher	Aroma	Dalcin	Brazil	2021	6.88
		Barth Haas	Germany	2020	4.50
Magnum	Bitter	Brava Terra	Brazil	2021	12.81
		Barth Haas	Germany	2020	14.70
Nugget	Bitter	Dalcin	Brazil	2021	9.66
		Barth Haas	United States	2018	9.50
Saaz	Aroma	Dalcin	Brazil	2021	5.67
		Barth Haas	Czech Republic	2020	3.50
Sorcachi Ace	Aroma/Bitter	Brava Terra	Brazil	2021	8.70
		Barth Haas	United States	2020	10.80

* The measurement of alpha acids in hops was conducted using high-performance liquid chromatography (HPLC). This method is regulated by organizations such as the American Society of Brewing Chemists (ASBC ^1^) and the European Brewery Convention (EBC ^2^). ^1^ ASBC methods of analysis—Hops-14 (HPLC method for alpha and beta acids in hops and hop products), American Society of Brewing Chemists, 2014. ^2^ Analytica-EBC, Method 7.4 (alpha acids in hops and hop products by HPLC), European Brewery Convention, 2010.

**Table 2 plants-13-02675-t002:** Compounds that showed their variation in expression (*p* ≤ 0.05) comparing the hop (Humulus lupulus) varieties planted in their country of origin with those planted in Brazil, in the hop variety of Hallertauer Mittelfruher, Magnum, Nugget, Saaz, and Sorachi Ace, planted in Germany (DE), United States (USA), Czech Republic (CZ), and Brazil (BR). Orange indicates higher expression, and blue is lower.

Compound	CAS #	Retention Index	Odor		Flavor	H. Mittelfrüher		Magnum		Nugget		Saaz		Sorachi Ace	
			Type	Strength	Type	DE	BR	DE	BR	USA	BR	CZ	BR	USA	BR
alpha-Pinene	80-56-8	948	Herbal	High	Woody										
2-Methyl-1-butanol	137-32-6	697	Ethereal	Medium	Ethereal										
2-Methyl-1-Butanol	1565-80-6	697													
beta-Pinene	127-91-3	943	Herbal	High	Pine										
Myrcene	123-35-3	958	Spicy	High	Woody										
alpha-Phellandrene	99-83-2	969	Terpenic	Medium	Terpenic										
alpha-Terpinene	99-86-5	998	Woody	Medium	Terpenic										
dextro-Limonene	5989-27-5	1018	Citrus	Medium	Citrus										
alpha-Terpineol	555-10-2	964	Minty	Medium											
Pentan-2-yl propanoate	54004-43-2	920													
Methyl (E)-4-methylpent-2-enoate	50652-78-3	828													
2-Pentene, 1-ethoxy-4-methyl-, (Z)-	51149-75-8	836													
2-Methylpropyl 3-methylbutanoate	589-59-3	955	Fruity	Medium	Green										
Isoamyl isobutyrate	2050-1-3	955													
2-Methylbutyl 2-methylpropanoate	2445-69-4	955	Fruity												
Ortho-cymene	527-84-4	1042													
1-Methyl-4-propan-2-ylidenecyclohexene	586-62-9	1052	Herbal	Medium	Woody										
Methyl heptanoate	106-73-0	984	Fruity		Fruity										
2-Octanone	111-13-7	952	Earthy	Medium	Dairy										
3-Methylbut-2-enyl 2-methylpropanoate	76649-23-5	1004	Fruity												
Methyl Methylenecyclohexanoate	73805-48-8	951													
1-Octen-3-ol	3391-86-4	969	Earthy	High	Mushroom										
3-Methylbutyl 2-methylbutanoate	27625-35-0	1054	Fruity		Fruity										
2-Methylbutyl 2-methylbutyrate	2445-78-5	1054	Fruity		Fruity										
Isoamyl isovalerate	2445-77-4	1054	Fruity		Fruity										
Methyl 6-methylheptanoate	2519-37-1	1019													
Perillene	539-52-6	1125	Woody	Medium											
Hexahydro-1,1-dimethyl-4-methylene-4H-cyclopenta[c]furan	344294-72-0	1052													
Hexyl 2-methylpropanoate	2349-7-7	1054													
Methyl octanoate	111-11-5	1083	Waxy		Green										
2-Methylpropyl hexanoate	105-79-3	1118	Fruity	Medium	Fruity										
Benzaldehyde	100-52-7	982	Fruity	High	Fruity										
2-Nonanone	821-55-6	1052	Fruity	Medium	Cheesy										
4-Hydroxy-3-hexanone	4984-85-4	916													
Octanol	111-87-5	1059	Waxy	Medium	Waxy										
Linalool	78-70-6	1082	Floral	Medium	Citrus										
Methyl 6-methyloctanoate	5129-62-4	1118													
Heptyl propanoate	2216-81-1	1183	Floral		Fruity										
Heptyl isobutyrate	2349-13-5	1218	Fruity		Berry										
Methyl nonanoate	1731-84-6	1183	Fruity		Winey										
Hexanoic acid	142-62-1	974	Fatty	Medium	Cheesy										
2-Methylbutyl hexanoate	2601-13-0	1218	Ethereal												
alpha-Ylangene	14912-44-8	1221													
alpha-Copaene	3856-25-5	1221	Woody												
2-Decanone	693-54-9	1151	Floral	Medium	Fermented										
Decyl trifluoroacetate	333-88-0	1216													
7-Decen-2-one	35194-33-3	1159													
Methylidenenonene	55050-40-3	1156	Aldehydic	Medium											
5-methylhexanoic acid	628-46-6	1009	Fatty	Medium											
2-(2,4-difluorophenyl)-1-[4-[6-(4-methylpiperazin-1-yl)pyridazin-3-yl]piperazin-1-yl]ethanone	1191-2-2	868													
beta-Copaene	18252-44-3	1216													
2-Undecanone	112-12-9	1251	Fruity	Medium	Waxy										
beta-Caryophyllene	87-44-5	1494	Spicy	Medium	Spicy										
(Z)-Undec-6-en-2-one	107853-70-3	1259													
Trans-geranic acid methyl ester	1189-9-9	1054													
(E)-beta-farnesene	18794-84-8	1440	Woody												
alpha-Humulene	6753-98-6	1579	Woody												
(Z,E)-alpha-Farnesene	26560-14-5	1458													
(-)-alpha-Muurolene	10208-80-7	1440	Woody												
alpha-Selinene	473-13-2	1474	Amber												
(3R,4aS,8aR)-8a-methyl-5-methylidene-3-prop-1-en-2-yl-1,2,3,4,4a,6,7,8-octahydronaphthalene	17066-67-0	1469	Herbal												
alpha-Farnesene	502-61-4	1458	Woody		Green										
(-)-alpha-Gurjunene	489-40-7	1419	Woody												
(-)-gamma-Cadinene	39029-41-9	1435	Woody	Medium											
(+)-delta-Cadinene	483-76-1	1469	Herbal												
Zonarene	41929-5-9	1440													
Naphthalene, 1,2,3,4,4a,7-hexahydro-1,6-dimethyl-4-(1-methylethyl)-	16728-99-7	1440													
alpha-Cedrene	24406-5-1	1440													
Perilla alcohol	18457-55-1	1261													
2-Tridecanone	593-8-8	1164													
Geranyl Propionate	105-90-8	1451	Floral	Medium	Waxy										
Calamenene	483-77-2	1537	Herbal Spicy	Medium											
Geranyl Butyrate	106-29-6	1550	Fruity	Medium	Fruity										
(Z,Z)-1,8,11-heptadecatriene	56134-3-3	1164													
(Z)-3-decen-1-yl acetate	81634-99-3	1389													
Methyl ester 3,6-dodecadienoic acid	16106-1-7	1164													
Linalool oxide	5989-33-3	1164	Earthy	Medium											
beta-Calacorene	50277-34-4	1542													
Heneicosapentaenoic acid methyl ester	65919-53-1	2415													
Linolenyl Alcohol	506-44-5	2077													
2-n-Butyl-2-cyclopentenone	5561_5-7	1280													
1-Epi-cubenol	19912-67-5	1580													
beta-Caryophyllene oxide	1139-30-6	1507	Woody	Medium	Woody										
Isoascaridole	19888-33-6	1592	Herbal												
Neointermedeol	5945-72-2	1613													
Humulene oxide II	19888-34-7	1592													
Methyl (8Z,11Z,14Z,17Z)-icosa-8,11,14,17-tetraenoate	132712-70-0	2308													
Cadalene	483-78-3	1706													
Caryophylla-4(12),8(13)-dien-5.alpha.-ol	19431-79-9	1677													
Methyl (Z)-5,11,14,17-eicosatetraenoate	59149-1-8	1280													

**Table 3 plants-13-02675-t003:** Unique volatile for each hop (*Humulus lupulus*) variety compared between samples planted and their country of origin with planted in Brazil. The varieties included Hallertauer Mittelfruher, Magnum, Nugget, Saaz, and Sorachi Ace, planted in Germany (DE), United States (USA), Czech Republic (CZ), and Brazil (BR). The green color indicates the presence of a specific compound in the hop samples studied.

Compound	CAS #	Retention Index	Odor		Flavor	H. Mittelfrüher		Magnum		Nugget		Saaz		Sorachi Ace	
			Type	Strength	Type	DE	BR	DE	BR	USA	BR	CZ	BR	USA	BR
2-Methyl-1-butanol	137-32-6	697	Ethereal	Medium	Ethereal										
Methyl isobutyrate	547-63-7	621	Fruity		Ethereal										
Dimethyl disulfide	624-92-0	722	Sulfurous		Sulfurous										
alpha-Phellandrene	99-83-2	969	Terpenic	Medium	Terpenic										
Isovaleric acid	503-74-2	811	Cheesy	High	Cheesy										
Heptyl isobutyrate	2349-13-5	1218	Fruity		Berry										
Linalool oxide	5989-33-3	1164	Earthy	Medium											
Acetone	67-64-1	455	Solvent	High											
Geranyl isovalerate	109-20-6	1586	Fruity	Medium	Green										
alpha-Myrcene	3338-55-4	976	Floral	Medium	Green										
Hexyl acetate	142-92-7	984	Fruity	Medium	Fruity										
Pentyl isobutyrate	2445-72-9	1019	Fruity												
2-Nonanol	628-99-9	1078	Waxy		Waxy										
(-)-gamma-Elemene	29873-99-2	1431	Green	Medium											
(-)-gamma-Cadinene	39029-41-9	1435	Woody	Medium											
beta-Cadinene	523-47-7	1440	Woody	Medium											
Neryl butyrate	999-40-6	1550	Green		Green										
Calamenene	483-77-2	1537	Herbal Spicy	Medium											
Isoascaridole	19888-33-6	1592	Herbal												
Cedrol	19435-97-3	1580	Herbal	Medium											
alpha-Muurolene	17699-14-8	1344	Herbal												
beta-Bisabolene	28973-97-9	1440	Green												
Methyl Isovalerate	23747-45-7	940	Cheesy		Fermented										
1-Hexanol	111-27-3	860	Herbal		Green										
Beta-Myrcene	502-99-8	958	Fruity	Medium											
Methyl Heptenone	110-93-0	938	Citrus	Medium	Green										
Isoamyl Isovalerate	2445-77-4	1054	Fruity		Fruity										
Methyl Nonenoate	13481-87-3	1191	Fruity	Medium	Fruity										
Delta-3-Carene	13474-59-4	1430	Woody												
Viridiflorol	20307-83-9	1446	Herbal	Medium											
Geranyl Propionate	105-90-8	1451	Floral	Medium	Waxy										
Geranyl Butyrate	106-29-6	1550	Fruity	Medium	Fruity										
Beta-Farnesene	21391-99-1	1547	Woody	Medium											
Muscone	37609-25-9	2072	Musk	Medium											
Alpha-Phellandrene	3779-61-1	976	Sweet Herbal	Medium											
Octyl Isobutyrate	109-15-9	1317	Waxy	Medium	Creamy										
Methyl 2-Methylbutanoate	868-57-5	721	Fruity		Fruity										
Methyl Isovalerate	556-24-1	721	Fruity	Medium	Fruity										
beta-Pinene	127-91-3	943	Herbal	High	Pine										
Methyl Caproate	106-70-7	884	Fruity	Medium	Fruity										
Ethyl Propanoate	105-37-3	686	Fruity	High	Fruity										
2-Methyl-1-Butanol	1565-80-6	697													
Isobutyl Isobutyrate	97-85-8	856	Fruity		Fruity										
Ethyl Isohexanoate	25415-67-2	920	Fruity												
alpha-Terpineol	555-10-2	964	Minty	Medium											
Dodecane	112-40-3	1200	Alkane												
Methyl Octanoate	15870-7-2	884													
Methyl 2-Methylheptanoate	51209-78-0	1019													
6-Methyl-5-Hepten-2-one	49852-35-9	896													
2-Octanone	111-13-7	952	Earthy	Medium	Dairy										
Methyl Methylenecyclohexanoate	73805-48-8	951													
Beta-Pinene	514-95-4	992													
Methyl 7-Methyloctanoate	2177-86-8	1118													
Octyl Acetate	112-14-1	1183	Floral	Medium	Waxy										
Pentyl Cyclohexadiene	56318-84-4	1143													
Hexanoic Acid	142-62-1	974	Fatty	Medium	Cheesy										
7-Decen-2-one	35194-33-3	1159													
Methylidenenonene	55050-40-3	1156	Aldehydic	Medium											
Octyl Propanoate	142-60-9	1282	Fruity		Estery										
Pentadecene	13360-61-7	1502													
beta-Cedrene	30021-74-0	1435	Woody												
alpha-Cedrene	24406-5-1	992													
Geranyl Isovalerate	51117-19-2	1586													
Linolenyl alcohol	506-44-5	2077													
Methyl 4-Methylpentanoate	2412-80-8	820	Fruity		Fruity										
Carene	3387-41-5	897	Woody		Woody										
Ethyl Hexanoate	123-66-0	984	Fruity	High	Fruity										
Pentyl Propanoate	624-54-4	984	Fruity		Fruity										
4-Pentenyl Butyrate	30563-31-6	1073													
5-Methylheptan-2-ol	54630-50-1	915													
Ethyl 5-Methylhexanoate	10236-10-9	1019													
Methyl (E)-Hept-2-enoate	22104-69-4	992													
(Z)-Hex-2-enyl Acetate	56922-75-9	992													
Ethyl Octanoate	106-32-1	1183	Waxy	Medium	Waxy										
beta-Bisabolene	495-61-4	1500	Balsamic												
1-Methyloctyl acetate	14936-66-4	1218													
2,3,5-Trithiahexane	42474-44-2	1072	Sulfurous												
Alloisolongifolene	87064-18-4	1390													
1-Tetradecene	1120-36-1	1403													
Ethyl trans-4-decenoate	76649-16-6	1389	Green	Medium	Fatty										
Neryl isobutyrate	2345-24-6	1486	Fruity	Medium	Fruity										
Methyl petroselinate	2777-58-4	2085													
Germacrene B	15423-57-1	1603	Woody												
1-Tridecene	2437-56-1	1304													
Agarospirol	1460-73-7	1598													
Muurola-4,10(14)-dien-1 beta-ol	257293-90-6	1586													
Ethyl octanoate	106-32-1	1183	Waxy	Medium	Waxy										
Trans-propionate 2-methyl-cyclohexanol	15287-79-3	1208													
2-Methylcyclohexyl butyrate	15287-80-6	1307													
3-Methylpentanoic acid	105-43-1	910	Animal	Medium	Sour										
cis-3-Hexenyl butyrate	16491-36-4	1191	Green	Medium	Green										
(2S,4S)-2,4-Dimethylhexanoic acid methyl ester	14251-45-7	955													
Ethyl heptanoate	106-30-9	1083	Fruity	Medium	Fruity										
Hexyl propanoate	2445-76-3	1083	Fruity												
Neo-alloocimene	7216-56-0	993													
Methyl non-4-enoate	20731-19-5	1191													
2-Methyl-6-methyleneocta-2,7-dien-4-ol	14434-41-4	1200													
Eremophilene	10219-75-7	1474													
Geranyl acetate	105-87-3	1352	Floral	Medium	Green										
Gamma-maalinene	20071-49-2	1398													
(-)-Aristolene	6831-16-9	1403													
2-Tetradecanone	2345-27-9	1549													
trans-Nerolidol	40716-66-3	1564	Floral	Low	Green										
1-cyclododecyl-ethanone	28925-0-0	955													
Linolenyl alcohol	506-44-5	2077													
alpha-Cadinol	481-34-5	1580	Herbal	Medium											
2,2,4,6,6-pentamethylheptane	13475-82-6	981													
Butyl nitrite	544-16-1	609													
1-Pentanol	71-41-0	761	Fermented		Fusel										
2-methylbutyl propanoate	2438-20-2	920	Fruity												
Benzaldehyde	100-52-7	982	Fruity	High	Fruity										
6-methylheptanoic acid	929-10-2	1109													
5-Nonenoic acid methyl ester	20731-20-8	1191													
Heptanoic acid	111-14-8	1073	Cheesy		Waxy										
Neryl isobutyrate	2345-24-6	1486	Fruity	Medium	Fruity										
Perillyl alcohol	536-59-4	1261	Green	Medium	Woody										
Isopentyl 8-methylnon-6-enoate	1215128-16-7	1559													
4,8,11,11-tetramethylbicyclo[7.2.0]undec-3-en-5-ol	913176-41-7	1677													
(6Z,9Z,12Z,15Z)-Methyl octadeca-6,9,12,15-tetraenoate	73097-0-4	943													
Bicyclo[3.1.1]hept-2-ene, 2,6-dimethyl-6-(4-methyl-3-pentenyl)-	17699-5-7	1474													
Methyl 5-methyl-2-hexenoate	68797-67-1	928													
4,6-Dimethyloctanoic acid	2553-96-0	1154													
1,3-Nonadiene	56700-77-7	914													
Camphene	79-92-5	943	Woody	Medium	Camphoreous										
alpha-Terpinene	99-86-5	998	Woody	Medium	Terpenic										
Borneol	464-45-9	1138	Balsamic	Medium	Camphoreous										
10-Epizonarene	41702-63-0	1469													
alpha-Farnesene	502-61-4	1458	Woody		Green										
(+)-delta-Cadinene	483-76-1	1469	Herbal												
Naphthalene, 1,2,3,4,4a,7-hexahydro-1,6-dimethyl-4-(1-methylethyl)-	16728-99-7	1440													
1-Octadecene	112-88-9	1801													
(2Z,6E)-Farnesol	3790-71-4	1710													
4-methyl-5-propylnonane	62185-55-1	1185													
3-Methylbut-2-enyl 2-methylpropanoate	76649-23-5	1004	Fruity												
2,6-dimethyl-1,3,5,7-octatetraene, E,E-	460-1-5	955													
Heptyl propanoate	2216-81-1	1183	Floral		Fruity										
alpha-Guaiene	3691_12-1	1054	Woody												
Methyl linolelaidate	2462-85-3	955													
Methyl dodecanoate	111-82-0	1481	Waxy	Medium	Waxy										
Methyl decanoate	110-42-9	1282	Fermented		Fatty										
2-methylpropyl propanoate	540-42-1	955	Fruity		Fruity										
Pentan-2-yl propanoate	54004-43-2	920													
3-methylbutyl 3-methylbutanoate	659-70-1	1054	Fruity	Medium	Green										
S-propyl hexanethioate	2432-78-2	1303													
(Z)-Undec-6-en-2-one	107853-70-3	1259													
(-)-alpha-Gurjunene	489-40-7	1419	Woody												
.alpha.-Maaliene	489-28-1	1403													
Methyl petroselinate	2777-58-4	2085													
Perilla alcohol	18457-55-1	1261													
Heneicosapentaenoic acid methyl ester	65919-53-1	2415													
2,2-dimethyldecane	17302-37-3	1130													
3-methylbut-2-enal	107-86-8	692	Fruity		Fruity										
3,5-dimethyl-1,6-heptadiene	68701-99-5	768													
(Z)-9-methylundec-5-ene	74630-65-2	1158													
Methyl (E)-4-methylpent-2-enoate	50652-78-3	828													
(Z)-hex-3-en-1-ol	928-96-1	868	Green	High	Green										
2-methylbutanoic acid	116-53-0	811	Acidic	Medium	Fruity										
Methyl 2,4-dimethylnonanoate	54889-61-1	1253													
2-methyl-6-methylene-2-octene	10054-9-8	1054													
3,3-dimethylcyclohexan-1-one	2979-19-3	1025													
Decyl trifluoroacetate	333-88-0	1216													
2-Nonanone	821-55-6	1052	Fruity	Medium	Cheesy										
Fenchol	1632-73-1	1138	Camphoreous	Medium	Camphoreous										
11,14-Eicosadienoic acid, methyl ester	2463_02-7	1054													
beta-Bisabolene	495-61-4	1500	Balsamic												
(E)-alpha-bisabolene	25532-79-0	1518													
3-(1,1-dimethylethyl)-2,5-furandione	18261-7-9	1054													
1-methyl-3-propan-2-ylbenzene	535-77-3	1042													
Methyl (Z)-octadec-9-enoate	112-62-9	2085	Mild fatty	Low											
trans-Furan linalool oxide	34995-77-2	1164	Floral	Medium											
3-methoxybutan-2-ol	53778-72-6	692													
alpha-Pinene	80-56-8	948	Herbal	High	Woody										
Butyl 2-methylpropanoate	97-87-0	920	Fruity		Fruity										
2-Methylpropyl 3-methylbutanoate	589-59-3	955	Fruity	Medium	Green										
Isoamyl isobutyrate	2050-1-3	955													
(Z)-3-hexen-1-yl acetate	3681-71-8	992	Green	High	Green										
Methyl (2S,4R)-2,4-dimethylheptanoate	18450-78-7	1054													
2-methylpent-4-en-1-ol	5673-98-3	786													
[(Z)-hex-3-enyl] 2-methylbutanoate	53398-85-9	1226	Green	Medium	Green										
Methyl 10-undecenoate	111-81-9	1371	Fatty		Waxy										
alpha-Selinene	473-13-2	1474	Amber												
2-Isopropenyl-4a,8-dimethyl-1,2,3,4,4a,5,6,7-octahydronaphthalene	103827-22-1	1502													
Geraniol	106-24-1	1228	Floral	Medium	Floral										
Methyl ester 3,6-dodecadienoic acid	16106-1-7	1164													
2-hexadecyloxirane	7390-81-0	1054													
Neryl propionate	105-91-9	1451	Fruity	Medium	Green										
Aplotaxene	10482-53-8	1725	Costus												
2-Nonadecanone	629-66-3	2046													
Methyl 7Z-hexadecenoate	56875-67-3	1886													
Methyl (8Z,11Z,14Z,17Z)-icosa-8,11,14,17-tetraenoate	132712-70-0	2308													
Prenol	556-82-1	746	Fruity		Fruity										
Isobutyl 2-methylbutyrate	2445-67-2	955	Fruity		Fruity										
Ortho-cymene	527-84-4	1042													
Methyl heptanoate	106-73-0	984	Fruity		Fruity										
2-Methylbutyl 2-methylbutyrate	2445-78-5	1054	Fruity		Fruity										
Heptyl ester acetic acid	112-6-1	1054													
hexyl 2-methylpropanoate	2349-7-7	1054													
Methyl octanoate	111-11-5	1083	Waxy		Green										
Mesityl oxide	141-79-7	739	Vegetable		Potato										
2-Heptanone	110-43-0	853	Cheesy	High	Cheesy										
3-Methylbutyl propanoate	105-68-0	920	Fruity	Medium	Fruity										
1-Dodecene	112-41-4	1204													
1-Cyclohexyl-2-buten-1-ol	79605-62-2	1249													
2-Methylpropyl hexanoate	105-79-3	1118	Fruity	Medium	Fruity										
1,1-Cyclohexanedimethanol	2658-60-8	1339													
2-Methyl-2-(4-methyl-3-pentenyl)-cyclopropanemethanol	98678-70-7	1280													
2-Methyl-2-pentenoic acid	16957-70-3	1054	Fruity	Medium											
(1S,4S,4aS)-1-Isopropyl-4,7-dimethyl-1,2,3,4,4a,5-hexahydronaphthalene	267665-20-3	1440													
Calarene	17334-55-3	1403													
Cyclodecene	3618-12-0	1181													
Alpha-dehydro-ar-himachalene	78204-62-3	1601													
2-ethylidene-1,7,7-trimethylbicyclo[2.2.1]heptane	62413-60-9	1134													
2-n-Butyl-2-cyclopentenone	5561_5-7	1054													
beta-Caryophyllene oxide	1139-30-6	1507	Woody	Medium	Woody										
(-)-Camphene	5794_04-7	1280													
5-Methylhexan-3-one	623-56-3	789													
2-Pentene, 1-ethoxy-4-methyl-, (Z)-	51149-75-8	836													
(E)-hex-4-en-1-ol	928-92-7	868	Green		Green										
Di(imidazol-1-yl)methanone	530-62-1	1445													
Ethyl 3-hexenoate	2396-83-0	992	Fruity	Medium	Fruity										
2-methylbutyl butanoate	51115-64-1	1019	Fruity		Fruity										
1-Heptanol	111-70-6	960	Green	Medium	Solvent										
Methyl nonanoate	1731-84-6	1183	Fruity		Winey										
2-Methylbutyl hexanoate	2601-13-0	1218	Ethereal												
2-methyl-6-methylideneocta-1,7-dien-3-one	41702-60-7	1076													
3,5-Dimethyl-1,6-heptadiene	68701-99-5	768													
Nalpha,Nomega-Dicarbobenzoxy-L-arginine	53934-75-1	3543													
2-(2,4-difluorophenyl)-1-[4-[6-(4-methylpiperazin-1-yl)pyridazin-3-yl]piperazin-1-yl]ethanone	1191-2-2	868													

## Data Availability

Data are contained within the article and Appendix A.

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
