# Peer review of "Hops across Continents: Exploring How Terroir Transforms the Aromatic Profiles of Five Hop (Humulus lupulus) Varieties Grown in Their Countries of Origin and in Brazil"

_plants, 2024, doi:10.3390/plants13192675_

Round 1
Reviewer 1 Report
Comments and Suggestions for Authors
In this paper, the content of volatile compounds in Humulus lupulus from different regions was extracted by HS-SPME/GC-MS, and it was found that there were significant differences in Humulus lupulus from different regions. However, this article has some minor problems, which are detailed below.
1. Since the paper mainly focuses on volatile oil, we should try to mention as much as possible in the third paragraph of the Introduction. The first time the abbreviation appears in the fourth paragraph write the full name (HS-SPME-GC-MS-O).
2. Delete lines 66-68.
3. Please mark clearly how the last ingredient content in Table 1 was measured or directly quoted.
4. In Section 2.2, make sure all numbers and units are spaced.
5. It is suggested that section 2.4 could be combined with section 2.3.
6. I didn't find Table 2 in the paper.
7. The classification of compounds is recommended in Tables 3 and 4, and the compounds appearing in sections 3.2-3.8 need to be labeled in which table.
8. Keep the same tense when describing the results, as in section 3.4. Also, 23 ingredients on line 253? Please check.
9. References 3 and 12 are inconsistent with the others, are the books? I suggest adding some more content to the conclusion.
Author Response
Dear Reviewers,
We would like to express our sincere gratitude for your thoughtful and detailed feedback on our manuscript. Your insights have been invaluable in improving the quality of our work. We have carefully addressed all your suggestions and made the necessary revisions throughout the manuscript.
To facilitate your review of the changes, we have highlighted the modifications in different colors: the revisions based on Reviewer 1's suggestions are marked in green, while those related to Reviewer 2's feedback are highlighted in yellow. We hope this will make it easier for you to see the adjustments we have made.
Thank you once again for your time and effort in reviewing our manuscript. We look forward to your feedback and hope the revised version meets your expectations.
Sincerely,
Marcos Edgar Herkenhoff
The questions and their respective responses are being sent separately to each reviewer and are outlined below:
Reviewer #1
In this paper, the content of volatile compounds in Humulus lupulus from different regions was extracted by HS-SPME/GC-MS, and it was found that there were significant differences in Humulus lupulus from different regions. However, this article has some minor problems, which are detailed below.
- Since the paper mainly focuses on volatile oil, we should try to mention as much as possible in the third paragraph of the Introduction. The first time the abbreviation appears in the fourth paragraph write the full name (HS-SPME-GC-MS-O).
Authors: We expanded the introduction and addressed some issues regarding volatile compounds. We believe this has addressed the reviewer's suggestions.
- Delete lines 66-68.
Authors: This detail went unnoticed by us. We have appropriately deleted those lines.
- Please mark clearly how the last ingredient content in Table 1 was measured or directly quoted.
Authors: The measurement of alpha acids in hops was performed using high-performance liquid chromatography (HPLC). This method is regulated by organizations such as the American Society of Brewing Chemists (ASBC) and the European Brewery Convention (EBC). This information has been added to the footer of the relevant table (Table 1).
- In Section 2.2, make sure all numbers and units are spaced.
Authors: All the numbers have been carefully reviewed and appropriately separated from their respective units.
- It is suggested that section 2.4 could be combined with section 2.3.
Authors: We agree with the reviewer and have combined these two sections.
- I didn't find Table 2 in the paper.
Authors: The issue with Table 2 is that it is too large to fit within the template suggested by the journal. However, to address this, we have decided to convert it into supplementary material; it is now referred to as Table S1.
- The classification of compounds is recommended in Tables 3 and 4, and the compounds appearing in sections 3.2-3.8 need to be labeled in which table.
Authors: The compounds from these tables have been labeled at the beginning of each paragraph to indicate in which table they are located.
- Keep the same tense when describing the results, as in section 3.4. Also, 23 ingredients on line 253? Please check.
Authors: The sentences have been revised, and we believe they are now improved.
- References 3 and 12 are inconsistent with the others, are the books? I suggest adding some more content to the conclusion.
Authors: In fact, reference number 3 pertains to a review article. We did not find any issues with it. You can verify it here: https://onlinelibrary.wiley.com/doi/full/10.1002/jib.160.
Regarding reference 12, it is indeed not an article, but it is also not a book. It is a webpage containing a conference summary. Regarding the conclusions, we chose to completely rewrite this section. We believe it has significantly improved.
Reviewer 2 Report
Comments and Suggestions for Authors
Manuscript ID plants-3210069
Lines 59-60 : "this study aimed to evaluate the aromatic profile of (several) hop varieties".
One can reasonably think that identical plants grown in different terroirs will produce significantly different plants, particularly in the case of volatile compounds. The text provides a qualitative characterization of this effect using colors: Tables 3 and 4. The goal is achieved.
Strengths
The presentation of Figure 1 is very relevant. By using a limited number of colors, Tables 3 and 4 qualitatively describe this effect. Furthermore, based on the list of observed compounds (Tables 3 and 4), a lengthy list of relevant references also supports the variations in pharmacological and chemical properties of the volatiles of different plants.
Recommendation
Regarding Tables 3 and 4, it is common to also find the retention indices for each compound. See the note at the end of this text.
Even if the authors in a previous publication (Ref. 20) mention the systematic names of the compounds, for easier reading of the tables, using common familiar names would be preferable to using systematic names. Adjustments to the text are also desirable: Lines 264, 293, 308, 378, 399, ... For the benefit of the authors, amended and abbreviated Tables 3 and 4 are included.
Details in text (errors – typos, …)
Lines 66-68 are unnecessary. The same applies to lines 492-493, 501-506 and 511-516.
Tables 3 and 4
While it is not impossible, the presence of fluorinated compounds is rather unlikely and highly questionable. Several CAS numbers are incorrect. Furthermore, the use of capital letters in compound names requires attention.
For example, if the correct CAS number for isobutyl hexanoate is 2349-07-7 rather than 2349_7-7, the compound can be found in the Perflavory database. Similarly, if the Number 2050_1-3 is actually 2050-01-3, the difluorinated compound becomes isobutyl isobutyrate. It is also found in the same database. See the previous comment.
List of references
It is generally complete. However, some desirable adjustments are noted.
1. The names of plants should be written in italics.
2. Some references are written with unnecessary capital letters. Example Ref. 18: Correlations between quality indicators of dry-aged beef and microbial succession during fermentation; Ref. 39, ...
3. The newspaper headlines are often reported in full, while others are abbreviated. Example Ref. 20 and 22 : Food Research International and Food Res Int;
4. The number of cited authors exceeds twenty in Ref. 29. Would it be preferable to limit the list to three or four authors for each reference?
5. For some references, the volume number is supplemented by the issue number. Example Ref. 4, Molecules, 2023, 28(16), 6100. For others, not. Example Ref. 16, Molecules, 2018, 23, 2866; Ref. 49, …
6. Few mistakes :
a. Ref. 5 : … Barriga, C.; Romero, E.; Cubero, J. … See comment No 4;
b. Ref. 33 : Food Chem X, 2024, 21, 101091
c. Ref. 34: … Mboussaah, A.D.K.; François Zambou N.F.,
d. Ref. 50: doi:10.1016/j.biombioe.2020.105919
Final note.
This note is reserved for the sole needs of the authors and should not be used to support the editor's decision.
According to the authors, chromatographic analysis was carried out on a polar column (Innowax). Indeed, the order of appearance (elution time) (Table 3) of the compounds is approximately in the correct order. At least one case requires further consideration. The α-gurjunene (489-40-7) should be located higher in this Table 3, in the vicinity of decan-2-one. It is also known that the mass spectrum of α-gurjunene is very similar to that of iso-α-gurjunene (73346-42-6), which should have a retention index more consistent with its position in Table 3. Could there be confusion?

Author Response
Dear Reviewers,
We would like to express our sincere gratitude for your thoughtful and detailed feedback on our manuscript. Your insights have been invaluable in improving the quality of our work. We have carefully addressed all your suggestions and made the necessary revisions throughout the manuscript.
To facilitate your review of the changes, we have highlighted the modifications in different colors: the revisions based on Reviewer 1's suggestions are marked in green, while those related to Reviewer 2's feedback are highlighted in yellow. We hope this will make it easier for you to see the adjustments we have made.
Thank you once again for your time and effort in reviewing our manuscript. We look forward to your feedback and hope the revised version meets your expectations.
Sincerely,
Marcos Edgar Herkenhoff
The questions and their respective responses are being sent separately to each reviewer and are outlined below:
Reviewer #2
Lines 59-60 : "this study aimed to evaluate the aromatic profile of (several) hop varieties".
One can reasonably think that identical plants grown in different terroirs will produce significantly different plants, particularly in the case of volatile compounds. The text provides a qualitative characterization of this effect using colors: Tables 3 and 4. The goal is achieved.
Strengths
The presentation of Figure 1 is very relevant. By using a limited number of colors, Tables 3 and 4 qualitatively describe this effect. Furthermore, based on the list of observed compounds (Tables 3 and 4), a lengthy list of relevant references also supports the variations in pharmacological and chemical properties of the volatiles of different plants.
Recommendation
Regarding Tables 3 and 4, it is common to also find the retention indices for each compound. See the note at the end of this text.
Authors: We agree and have added this information to these two tables, which have been renamed as Table 2 and Table 3, respectively.
Even if the authors in a previous publication (Ref. 20) mention the systematic names of the compounds, for easier reading of the tables, using common familiar names would be preferable to using systematic names. Adjustments to the text are also desirable: Lines 264, 293, 308, 378, 399, ... For the benefit of the authors, amended and abbreviated Tables 3 and 4 are included.
Authors: All compounds, in addition to being updated in the tables, have also been updated in the text.
Details in text (errors – typos, …)
Lines 66-68 are unnecessary. The same applies to lines 492-493, 501-506 and 511-516.
Authors: This detail went unnoticed by us. We have appropriately deleted those lines. Regarding the section between lines 511-516, we cannot delete this part about the authors' contributions, as it is a requirement of the journal. We must adhere to this format.
Tables 3 and 4
While it is not impossible, the presence of fluorinated compounds is rather unlikely and highly questionable. Several CAS numbers are incorrect. Furthermore, the use of capital letters in compound names requires attention.
For example, if the correct CAS number for isobutyl hexanoate is 2349-07-7 rather than 2349_7-7, the compound can be found in the Perflavory database. Similarly, if the Number 2050_1-3 is actually 2050-01-3, the difluorinated compound becomes isobutyl isobutyrate. It is also found in the same database. See the previous comment.
Authors: We agree and have made the suggested modifications. Regarding the use of capital letters, we have now decided to follow a consistent format. All compounds will begin with capital letters, except for those starting with a written Greek letter, such as alpha or beta. The CAS numbers have also been corrected.
List of references
It is generally complete. However, some desirable adjustments are noted.
- The names of plants should be written in italics.
- Some references are written with unnecessary capital letters. Example Ref. 18: Correlations between quality indicators of dry-aged beef and microbial succession during fermentation; Ref. 39, ...
- The newspaper headlines are often reported in full, while others are abbreviated. Example Ref. 20 and 22 : Food Research International and Food Res Int;
- The number of cited authors exceeds twenty in Ref. 29. Would it be preferable to limit the list to three or four authors for each reference?
- For some references, the volume number is supplemented by the issue number. Example Ref. 4, Molecules, 2023, 28(16), 6100. For others, not. Example Ref. 16, Molecules, 2018, 23, 2866; Ref. 49, …
- Few mistakes :
- Ref. 5 : … Barriga, C.; Romero, E.; Cubero, J. … See comment No 4;
- Ref. 33 : Food Chem X, 2024, 21, 101091
- Ref. 34: … Mboussaah, A.D.K.; François Zambou N.F.,
- Ref. 50: doi:10.1016/j.biombioe.2020.105919
Authors: We have made the modifications suggested by the reviewer and conducted a thorough evaluation of each reference. All changes are highlighted in yellow. Regarding the excess of authors, we did not find any specific rules on this matter. Therefore, we chose to leave all the authors listed and allow the editors to make that decision.
Final note.
This note is reserved for the sole needs of the authors and should not be used to support the editor's decision.
According to the authors, chromatographic analysis was carried out on a polar column (Innowax). Indeed, the order of appearance (elution time) (Table 3) of the compounds is approximately in the correct order. At least one case requires further consideration. The α-gurjunene (489-40-7) should be located higher in this Table 3, in the vicinity of decan-2-one. It is also known that the mass spectrum of α-gurjunene is very similar to that of iso-α-gurjunene (73346-42-6), which should have a retention index more consistent with its position in Table 3. Could there be confusion?
Authors: It may seem a bit unusual, but we chose to organize the compounds according to their presence in the sample order, or by their expression levels, from higher to lower. Otherwise, using colors in the tables wouldn't make much sense. Our intention was to make it easier for the reader to understand which compounds differed between the groups.
Reviewer 3 Report
Comments and Suggestions for Authors
The manuscript entitled "Hops Across Continents: Exploring How Terroir Transforms the Aromatic Profiles of Five Hop (Humulus lupulus) Varieties Grown in Their Countries of Origin and in Brazil" by Herkenhoff et al., reported natural variations in aroma profile of Hop varieties in Brazil and the country of their origin by using HS-SPME/GC-MS methodology. The study revealed the impact of terroir on the aromatic profile of hop varieties, and most the Hops varieties show distinct aromatic profiles depending on their cultivation geographs, whether they are grown in Brazil or their country of origin (Germany). The aroma metabolite profiling of four various Hops varieties cultivated in different terroirs revealed the impacts of geographic and climatic influences on Hops Metabolites, which will be potentially helpful to their agricultural practices. The English writing and data presentation are clear and careful, metabolite profiling experiments with HS-SPME/GC-MS were well-designed, and controled in methodology and data treatments.
Comments on the Quality of English Language
no